# Mapping mechanical stress in curved epithelia of designed size and shape

Ariadna Marín-Llauradó[1,11], Sohan Kale [2,3,11] ✉, Adam Ouzeri[4], Tom Golde [1], Raimon Sunyer[1,5,6,7], Alejandro Torres-Sánchez[1,4,8], Ernest Latorre[1], Manuel Gómez-González [1], Pere Roca-Cusachs [1,5], Marino Arroyo [1,4,9] ✉ & Xavier Trepat [1,5,7,10] ✉

The function of organs such as lungs, kidneys and mammary glands relies on the three-dimensional geometry of their epithelium. To adopt shapes such as spheres, tubes and ellipsoids, epithelia generate mechanical stresses that are generally unknown. Here we engineer curved epithelial monolayers of controlled size and shape and map their state of stress. We design pressurized epithelia with circular, rectangular and ellipsoidal footprints. We develop a computational method, called curved monolayer stress microscopy, to map the stress tensor in these epithelia. This method establishes a correspondence between epithelial shape and mechanical stress without assumptions of material properties. In epithelia with spherical geometry we show that stress weakly increases with areal strain in a size-independent manner. In epithelia with rectangular and ellipsoidal cross-section we find pronounced stress anisotropies that impact cell alignment. Our approach enables a systematic study of how geometry and stress influence epithelial fate and function in three-dimensions.

The internal and external surfaces of the animal body are lined by thin cellular layers called epithelia. Epithelia are active materials that sculpt the early embryo, separate body compartments, protect against pathogenic and physicochemical attacks, and control fluid and biomolecular transport[1]. To perform these functions, most epithelia fold into three-dimensional structures that enclose a pressurized fluid-filled cavity called lumen. Some folded epithelia such as the trophectoderm or alveoli are nearly spherical[2]. Others, such as those lining nephrons or blood vessels are tubular[3,4]. Yet others, like the developing zebrafish otic vesicle[5] or the early *drosophila* embryo[6], are ellipsoidal. In general, epithelia display a combination of shapes and a broad diversity of sizes, with lumens ranging from a few microns to several millimeters[7–9]. These diverse geometries enable optimal physiological processes and influence cellular fate and function[10–13].

To adopt their three-dimensional geometry, epithelia generate active mechanical stresses. With the exception of purely spherical epithelia[8,14,15], no current technology enables mapping these stresses in 3D in absolute quantitative terms. To fill this gap, here we present an experimental and computational approach to design epithelia of controlled geometry and to map the stress tensor everywhere in the monolayer without assumptions of mechanical properties. Using this approach, we show that the relationship

---

[1]Institute for Bioengineering of Catalonia (IBEC), The Barcelona Institute for Science and Technology (BIST), 08028 Barcelona, Spain. [2]Department of Mechanical Engineering, Virginia Polytechnic Institute and State University, Blacksburg, VA 24061, USA. [3]Center for Soft Matter and Biological Physics, Virginia Polytechnic Institute and State University, Blacksburg, VA 24061, USA. [4]LaCàN, Universitat Politècnica de Catalunya-BarcelonaTech, Barcelona, Spain. [5]Facultat de Medicina, Universitat de Barcelona, 08036 Barcelona, Spain. [6]Institute of Nanoscience and Nanotechnology (IN2UB), Universitat de Barcelona, Barcelona, Spain. [7]Centro de Investigación Biomédica en Red en Bioingeniería, Biomateriales y Nanomedicina (CIBER-BBN), 08028 Barcelona, Spain. [8]European Molecular Biology Laboratory (EMBL) Barcelona, 08003 Barcelona, Spain. [9]Centre Internacional de Mètodes Numèrics en Enginyeria (CIMNE), 08034 Barcelona, Spain. [10]Institució Catalana de Recerca i Estudis Avançats (ICREA), Barcelona, Spain. [11]These authors contributed equally: Ariadna Marín-Llauradó, Sohan Kale. ✉e-mail: kale@vt.edu; marino.arroyo@upc.edu; xtrepat@ibecbarcelona.eu

between epithelial tension and strain is largely independent of lumen size. By engineering elliptical and tubular epithelia, we examine the link between the anisotropic stress tensor, cell shape, and cellular tractions.

## Results

To design curved epithelia with controlled geometry, we photo-patterned soft PDMS (3 kPa Young's Modulus) substrates with low fibronectin density motifs surrounded by high fibronectin density areas (Fig. 1a and Supplementary Fig. 1). MDCK cells attached on both high and low fibronectin density areas and formed a flat cohesive monolayer. After 24–48 h, the monolayer delaminated spontaneously from the low-density motifs to form a fluid-filled lumen. The basal geometry of the lumens closely followed the micropatterned motif, which we hereafter refer to as footprint (Fig. 1b, c). Lumen formation and inflation is driven by the well-known ability of MDCK cells to pump osmolytes in the apico-basal direction[16,17], which builds up sufficient osmotic pressure to delaminate cells from the low fibronectin motifs but not from the surrounding high fibronectin areas. This method allowed us to engineer precise lumens with a broader range of sizes and shapes than previous approaches[14].

Using this technique, we first investigated the mechanics of epithelial monolayers with circular footprints of diameters 25 μm, 50 μm, 100 μm, and 200 μm. In every case, the epithelial monolayer adopted a dome-like morphology that was well fitted by a spherical cap (Fig. 1b, c). Cell density on the domes did not vary with footprint size (Supplementary Fig. 2). We used traction microscopy to map the three-dimensional traction vectors at the substrate surface. Tractions under the suspended dome, which pointed uniformly towards the substrate, are a direct readout of luminal pressure, $\Delta P$. Pressure was balanced by an upwards traction at the contact point between the first ring of cells and the substrate. This out-of-plane traction was not purely tangential to the suspended dome (Fig. 1c), indicating a contribution of the adherent monolayer to the mechanical equilibrium at the contact point.

Thanks to the spontaneous fluctuations in dome volume (Supplementary Fig. 3), we were able to measure luminal pressure for epithelial curvatures spanning more than one order of magnitude (Fig. 1d). Pressure increased linearly with curvature and then tended to plateau, indicating a limit in the pressure that MDCK monolayers spontaneously build up. We next sought to infer the epithelial stress on the dome monolayer from tissue shape and luminal pressure. To do so, we assumed a membrane state of stress in the monolayer characterized by a symmetric 2 × 2 tensor, $\sigma$[14]. In spherically symmetric monolayers as domes, cysts, and blastocysts, mechanical equilibrium tangential to the tissue requires that stress is uniform and isotropic[18] and, therefore, $\sigma$ is diagonal and has equal diagonal elements $\sigma$, which correspond to the surface tension. Mechanical equilibrium normal to the tissue results in Young-Laplace's law, $\sigma = \frac{\Delta P \cdot R}{2}$, where $R$ is taken as the radius at half thickness of the monolayer, from which $\sigma$ can be computed.

We noticed that the contact angle between adjacent cells is smaller on the apical cellular surface than on the basal one, which indicates higher basal than apical surface tension (Supplementary Fig. 4). This observation prompted us to re-examine our membrane assumption. Indeed, apicobasal differences in tension could give rise to a self-generated bending moment or spontaneous curvature with a potentially relevant contribution to the mechanical balance of lumen pressure beyond Young-Laplace's law[19–22]. This effect should be size-dependent since spontaneous curvature introduces a length-scale. To answer this question, we quantified with computational 3D vertex models of domes the role of apicobasal asymmetry on inferred stresses using Young-Laplace's law, finding a negligible effect irrespective of dome size and for 9-fold differences in surface tension asymmetry (Supplementary Note 1 and Supplementary Fig. 12). These results

support the membrane assumption in the context of stress inference for tense epithelial domes.

We then studied how $\sigma$ varies with areal strain $\varepsilon_a$, defined as the change in tissue area normalized by the area of the footprint, for each footprint size, focusing on the range $\varepsilon_a < 100\%$ (Fig. 1e). Data were highly scattered, with comparable values for mean and standard deviation. Upon averaging over several domes and time points, we observed that $\sigma$ weakly increases with $\varepsilon_a$, consistent with the low-strain behavior we had previously defined as active superelasticity[14]. Remarkably, the stress-strain relationship did not depend on the footprint size. This result indicates that for a broad range of sizes, curvature does not trigger mechanosensing feedback loops that impact the magnitude of epithelial tension significantly.

Whereas Young-Laplace's law provides an exact expression of stress in spherical monolayer membranes, epithelia generally deviate from perfectly spherical geometries. We thus sought to develop a general formalism to map the full stress tensor in monolayers of arbitrary size and shape. This approach, which we call curved Mono-layer Stress Microscopy (cMSM), accounts for the two tangential equilibrium equations used in standard (planar) stress inference methods[23–28] and further exploits the equation of out-of-plane force balance available in the presence of curvature (Supplementary Fig. 5). Thus, unlike planar stress inference, with an equal number of equations and unknowns it is possible in principle to determine the three independent components of the stress tensor without any assumptions about material behavior other than a membrane state of stress. The formulation of cMSM, its finite element implementation, the criteria to select regularization parameters, and a thorough verification of the method is available in Supplementary Note 2.

We applied this force inference method to domes that deviate from a spherical cap. First, we photopatterned substrates with rectangular footprints of same area but different ratios between long and short axis lengths, ranging from 1 (square) to 4 (Fig. 2). Like for spherical domes, epithelia delaminated from the footprints and formed caps and tubes with slowly fluctuating levels of inflation (Fig. 2a). We then inferred the stress tensor from the measured luminal pressure and monolayer geometry. To visualize it, we computed the mutually orthogonal principal directions of stress and the principal stresses $\sigma_I$ and $\sigma_{II}$ along these directions. We plotted the hydrostatic or mean surface tension, $\sigma_I + \sigma_{II}$, as a colormap and the principal stresses as converging (negative) or diverging (positive) pairs of arrows along their corresponding directions. The tension maps inferred by cMSM exhibited spatial variability associated with specific geometric features of dome surfaces, but also systematic patterns such as markedly uniaxial stresses parallel to the short axis of the dome (Fig. 2b).

To validate cMSM, we reasoned that the traction component normal to the substrate at the contact line between the suspended monolayer and the substrate, $T_z$, should match closely the normal traction computed from the inferred stress on the suspended monolayer at that point (Fig. 2c). We tested these predictions as a function of epithelial inflation and footprint aspect ratio (Fig. 2d, e). Note that $\sigma$ has units of tension and thus needs to be divided by a length $l_T$ to compare it with $T_z$. This length should be understood as the thickness of the band lining the footprint over which tension in the suspended epithelium is transmitted to the substrate. We found $l_T = 15.7$ μm ("Methods" section), of the order of one cell diameter. As expected, the experimental values of $T_z$ were equal on all sides of the squares and increased with inflation (Fig. 2d), closely matching the inferred tractions (Fig. 2e). For rectangles, we found two determinants that produced significant differences in $T_z$. The first one was inflation, which increased tractions on each side of the rectangular footprint and for each aspect ratio probed (Fig. 2d). The second one was the length of the footprint side: $T_z$ was higher along the longer side of the footprint than along the shorter one (Fig. 2d). These trends are captured by the inferred traction map (Fig. 2e), particularly for high inflation. Together,

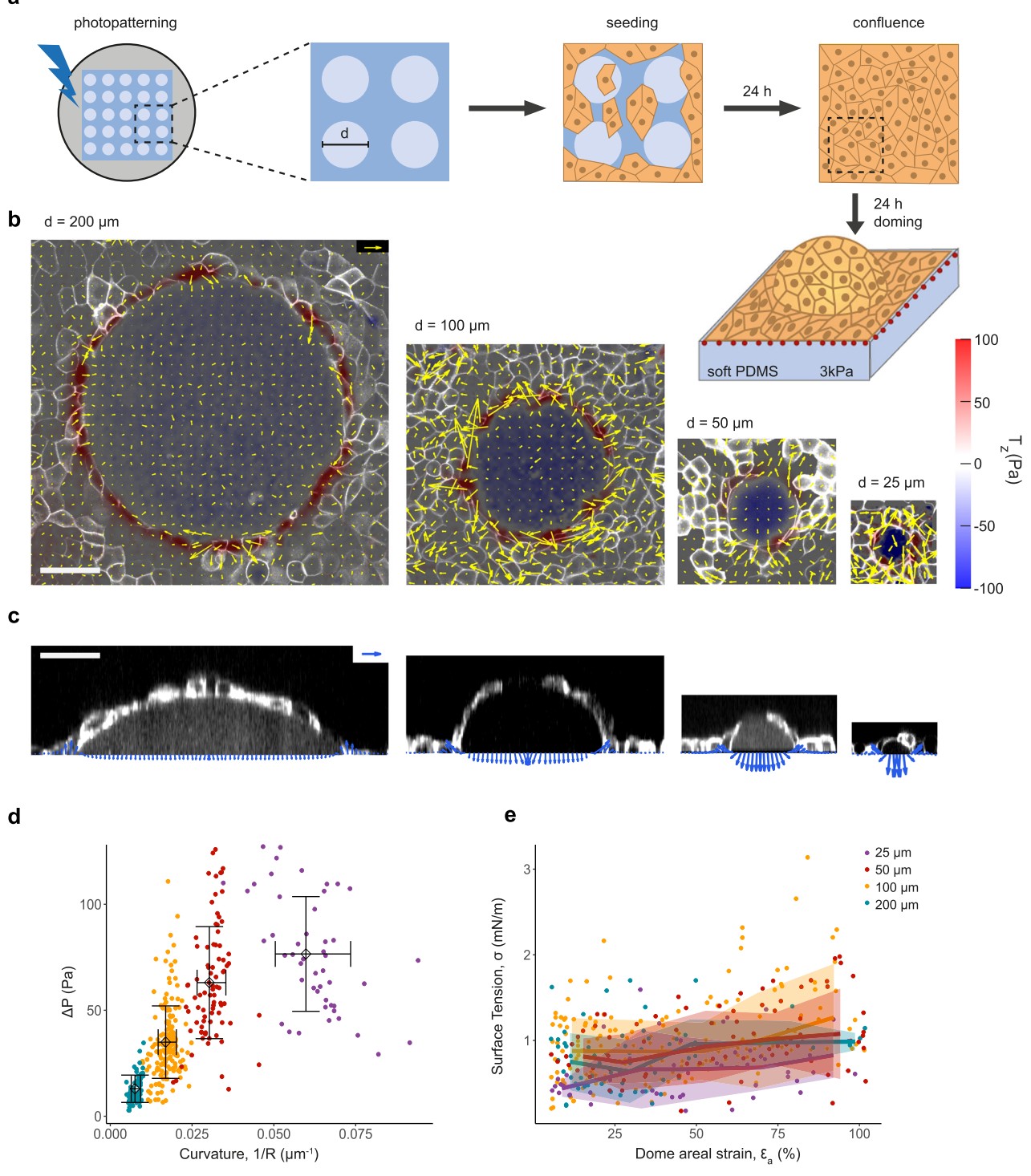

**Fig. 1 | Scaling of tension and pressure in spherical domes. a** Scheme of substrate photopatterning, cell seeding and dome formation. **b** 3D traction maps overlaid on top views of representative MDCK domes of 200 μm, 100 μm, 50 μm, and 25 μm in footprint diameter (from left to right). Yellow vectors represent in-plane horizontal components $T_x$ and $T_y$ and the color map represents the vertical component $T_z$. **c** Tractions overlaid on lateral views of the domes shown in b. Tractions were averaged circumferentially for plotting. In b,c data are representative of $n = 12$ domes (25 μm), 11 domes (50 μm), 17 domes (100 μm), and 13 domes (200 μm). Scale bars in b,c, 50μm. Scale vectors in b,c, 100 Pa. **d**, Dome

pressure as a function of dome curvature. Data shown as mean ± SD of $n = 12$ domes (25 μm), 11 domes (50 μm), 17 domes (100 μm), and 13 domes (200 μm) at different levels of inflation. Color coding indicates the footprint diameter (see legend in panel **e**). **e** Surface tension in the free-standing sheet as a function of nominal areal strain of the dome. The line and shaded area indicate mean ± SD by binning the data (equally spaced bins with $n \geq 3$ points per bin). Number of domes is the same as in panel **d**. Surface tension was computed with Laplace's law using the measured pressure and dome radius. Source data are provided as a Source Data file.

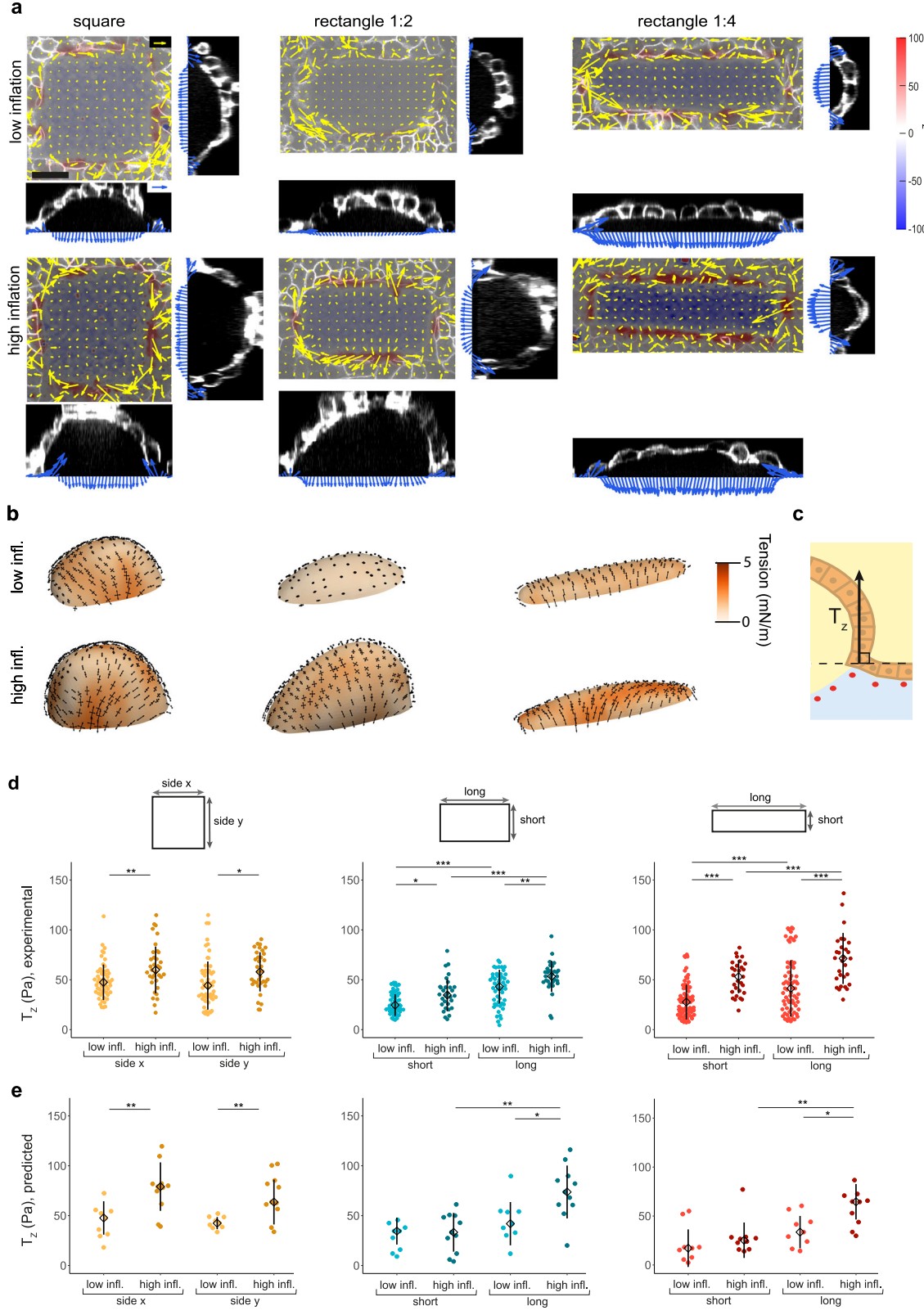

these data establish a close agreement between the inferred and measured stress at the contact line between the dome and the substrate, supporting the validity of our stress inference method.

To study systematically the effect of curvature on inferred stress we designed epithelia with an elliptical footprint and two different eccentricities (aspect ratios 1:3 and 2:3, Fig. 3a). In agreement with domes with rectangular footprints, stress was anisotropic, with the

maximum principal stress direction predominantly parallel to the short axis of the ellipse (Fig. 3b, see Supplementary Fig. 6 for additional domes). We then measured $T_z$ as a function of the polar angle along the elliptical footprint, $\beta$. For shallow domes, $T_z$ slightly depended on $\beta$ (Fig. 3c). As inflation levels grew, $T_z$ increased and became progressively anisotropic, with higher values of traction along the less curved regions of the ellipse. This behavior was more pronounced in domes of

**Fig. 2 | Stress in domes with squared and rectangular cross-section. a** Traction maps of rectangular domes with different aspect ratios (from left to right, 1:1, 1:2, and 1:4 aspect ratios) and inflation levels (low inflation, top row; high inflation, bottom row). Each one of the six panels comprises a top view (center) and two lateral views (right and bottom). The top view shows a 3D traction map, where yellow vectors represent the in plane ($T_x$, $T_y$) components and the color map represents the vertical component $T_z$. The blue vectors in the lateral views show the tractions along two orthogonal bands passing through the center of the footprint. Scale bar, 25 μm. Scale vector, 50 Pa. **b** Inferred stress tensor on the free-standing cell layer of the domes shown in **a**. The black arrows represent the two principal components of the inferred stress. The colormap represents the mean surface tension, $\sigma_I + \sigma_{II}$. **c** Schematic of the vertical traction component $T_z$ at the cell-substrate interface. **d** Experimental vertical tractions at the short and long sides of domes with low and high inflation levels. The tractions were averaged along the central 50% of the dome sides. Data are shown as median ± SD of $n = 53$ (square low), 35 (square high), 50 (rect. 1:2 low), 30 (rect. 1:2 high), 69 (rect. 1:4 low), and 29 (rect. 1:4 high). **e** Vertical tractions at the short and long sides of domes with low and high inflation levels obtained with cMSM. The tractions were averaged along the central 50% of the dome sides. Data are shown as median ± SD of $n = 9$ (square low),

11 (square high), 8 (rect. 1:2 low), 12 (rect. 1:2 high), 9 (rect. 1:4 low), and 10 (rect. 1:4 high). Statistical significance in **d**, **e** was determined using the two-sided Wilcoxon rank sum test for paired (when comparing side x and side y of the same domes) and unpaired (when comparing low inflation with high inflation domes) samples. Only statistically significant pairwise comparisons are indicated. *$P < 0.05$, **$P < 0.01$, and ***$P < 0.001$. **d** $P = 0.0032$ (left panel, low infl. side x vs high infl. side x), $P = 0.012$ (left panel, low infl. side y vs high infl. side y), $P = 1.46E\text{-}08$ (center panel, low infl. short vs low infl. long), $P = 4.66E\text{-}08$ (center panel, high infl. short vs high infl. long), $P = 0.011$ (center panel, low infl. short vs high infl. short), $P = 2.82E\text{-}03$ (center panel, low infl. long vs high infl. long), $P = 3.50E\text{-}10$ (right panel, low infl. short vs low infl. long), $P = 2.61E\text{-}07$ (right panel, high infl. short vs high infl. long), $P = 5.6E\text{-}07$ (right panel, low infl. short vs high infl. short), and $P = 1.37E\text{-}04$ (right panel, low infl. long vs high infl. long). **e** $P = 2.29E\text{-}03$ (left panel, low infl. side x vs high infl. side x), $P = 7.44E\text{-}03$ (left panel, low infl. side y vs high infl. side y), $P = 2.00E\text{-}03$ (center panel, high infl. short vs high infl. long), $P = 0.016$ (center panel, low infl. long vs high infl. long), $P = 3.91E\text{-}03$ (right panel, high infl. short vs high infl. long), and $P = 0.010$ (right panel, low infl. long vs high infl. long). Source data are provided as a Source Data file.

higher eccentricity. We next compared these experimental values with those predicted by our force inference method, using the experimentally measured geometry and luminal pressure as the sole input variables. cMSM predictions matched closely the experimental $T_z$, capturing the magnitude as well as its dependence on dome inflation, footprint eccentricity and polar angle (Fig. 3d). As in the case of rectangular footprints, the agreement was more pronounced at higher levels of inflation, where the assumption of membrane stress was more robust and the estimation of dome shape was more precise. To further validate our method, we treated elliptical the domes with a ROCK inhibitor Y27632 (Supplementary Fig. 7a–g). This treatment led to a rapid decrease in in-plane tractions away from the domes, indicating a loss in actomyosin activity (Supplementary Fig. 7h). As expected from such a loss in activity, Y27632 caused a rapid decrease in dome tension with little changes in dome shape, resulting in a consequent drop in pressure (Supplementary Fig. 7a–f). The rapid change in tension but not in shape or volume (Supplementary Fig. 7g) indicates that the observed rapid response to Y27632 stems from its role as a ROCK inhibitor rather than from its effect on atypical protein kinase C (aPKC) and on its downstream role on the maintenance of polarized fluid transport in MDCK cells.

We finally asked whether the inferred stress was predictive of cell orientation, as previously observed in 2D[23,29,30]. We segmented cells in the domes and fitted the shape of each cell to a 2D ellipse on the dome plane (Fig. 4a). We then computed the angle $\alpha$ between the longest axis of the ellipse and the maximum principal stress direction (Fig. 4b) and plotted the distribution of $\alpha$ for domes of high and low eccentricity. Data were binned according to the position of the cell center in the dome (top vs side and major vs minor axis regions, Fig. 4c, d). Both for high and low eccentricity, the angular distributions in the minor axis region (top and side) and in the major axis top region were skewed towards small angles, indicating a predominant alignment between the direction of maximum principal stress and cell elongation. This alignment was not stronger for higher magnitudes of the maximum stress or for increased stress anisotropy (Supplementary Fig. 8). Interestingly, the angular distributions of cells located in the major axis side region were skewed towards high angles. In these regions, maximum stress tended to be normal to the major axis, whereas cells tended to be parallel to it (Fig. 4a, c). These data suggest an additional competing mechanism for cell alignment. We propose that this mechanism might be associated with the cost of cell bending in the regions of highest curvature, as has been proposed for cells adhering on cylindrical wires[31]. If this mechanism dominates, cells will align in the direction of minimal curvature rather than in that of maximum stress to minimize bending. To test this idea, we measured curvature anisotropy across

the dome surface, defined as $1\text{-}k_{min}/k_{max}$, where $k_{min}$ and $k_{max}$ are the minimum and maximum curvatures, respectively (Supplementary Fig. 9). Consistent with our hypothesis, we found that in regions of relatively low curvature anisotropy (those labeled as minor axis top, minor axis side and major axis top on Fig. 4d), cells tend to align in the direction of maximum stress whereas in those of highest curvature anisotropy (major axis side region) they tend to align in the direction of minimal curvature. Together, these data support that a competition between stretching and bending determines cell orientation on curved surfaces.

## Discussion

The past decade has seen the development of numerous techniques to measure mechanical stresses in epithelial tissues including laser ablation[32], FRET tension sensors[33], droplet inserts[34], monolayer stress microscopy[23], cantilevers[35], and stress inference methods[36]. Whereas each of these techniques has advantages and limitations[37], none of them enables mapping the stress tensor in curved epithelia in absolute quantitative terms. Here we filled this gap by combining experimental and computational tools to design epithelia of arbitrary geometry and to map their luminal pressure and stress tensor. As a proof of concept, we used MDCK cell domes of spherical footprint to show that the relationship between stress and strain is largely independent of epithelial curvature, indicating that within the range of lumen size considered in this study, curvature is not mechanotransduced into changes in tension[38,39]. Using epithelia with elliptical footprint, we showed that cells tend to align with the direction of maximum principal stress, as previously observed in 2D monolayers[23,29,30], but this alignment was not universal and depended on geometry.

To map stresses on surfaces of arbitrary geometry, we developed a force inference method based solely on the measured geometry of the epithelium and luminal pressure. The method, which we call cMSM, uses the two tangential equilibrium equations (as in conventional 2D force inference[23–28]) and the normal equilibrium equation (à la Young-Laplace) to infer the three independent components of the epithelial stress tensor. Hence, unlike in planar stress inference, it does not require knowledge of the material properties of the monolayer. The only assumption of our approach is that the epithelium behaves as a 2D membrane, i.e. that it supports a two-dimensional state of stress. We tested the validity of this core assumption of the method using a 3D vertex model, which showed that a large apico-basal asymmetry has negligible effect on the inferred monolayer stress. In principle, in highly columnar and polarized monolayers, bending moments could play a stronger effect, but this situation is rare in epithelia surrounding pressurized lumens. In cMSM, epithelial shape allows us to infer the

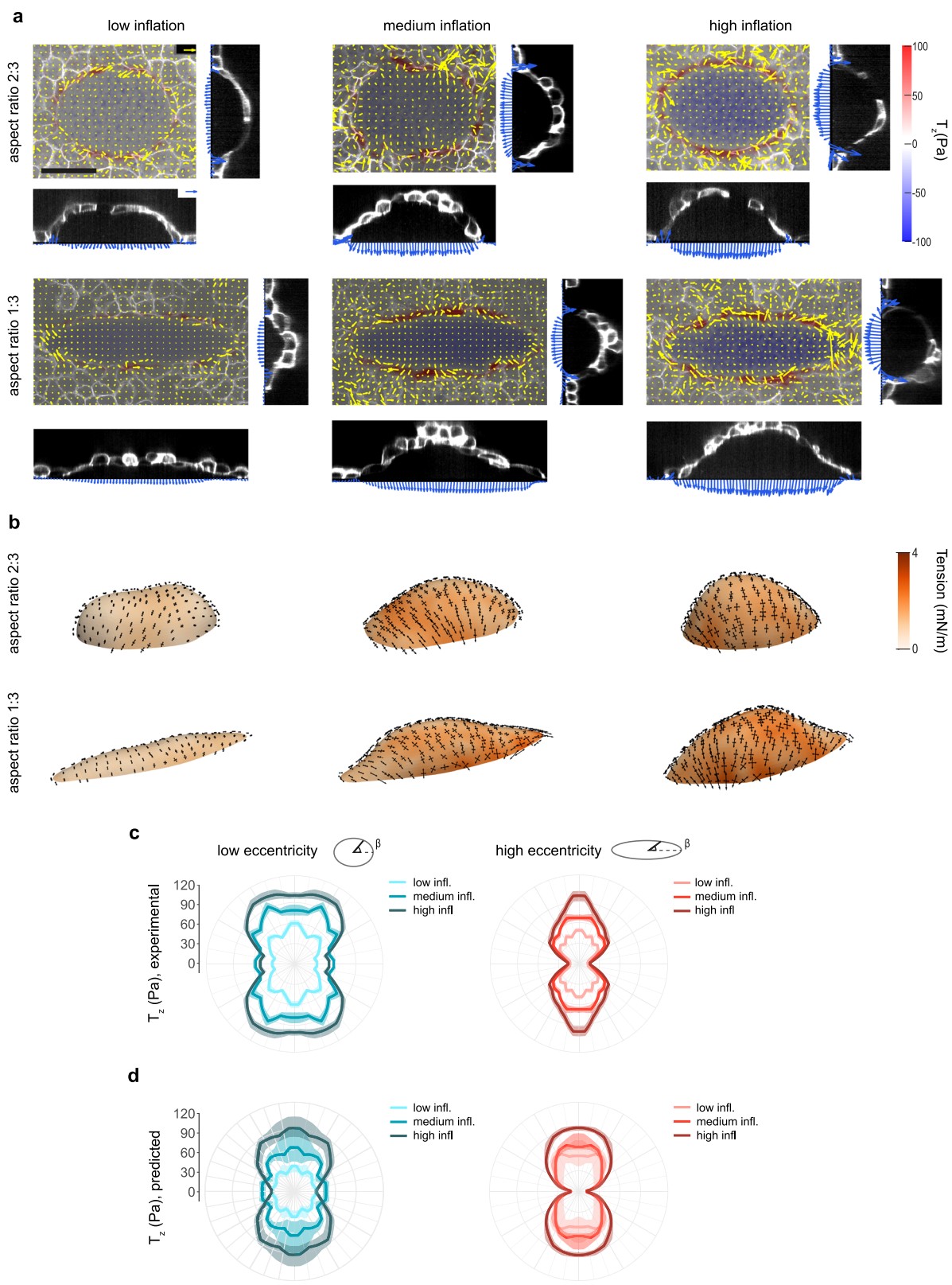

stress pattern whereas pressure provides the stress scale. Hence, in experimental conditions where both epithelial shape and pressure can be measured, as in the present study, cMSM enables an absolute quantification of the stress tensor. In situations where geometry but not pressure can be measured, as is often the case in vitro and in vivo, cMSM can still provide quantification of epithelial stress up to a scaling factor. Therefore, our technique is of general applicability beyond the

highly controlled conditions reported in this study (see also discussion in Supplementary Note 2).

To validate our technique, we designed tubular caps of different aspect ratio and elliptical caps of different eccentricity. In these geometries, the stress field inferred by cMSM was anisotropic, with higher values in the direction parallel to the short axis of the tubes and ellipsoids. These inferred anisotropic stresses were in close agreement

**Fig. 3 | Stress in ellipsoidal domes. a** Traction maps of ellipsoidal domes with different aspect ratios (2:3 top and 1:3 bottom) and inflation levels (from left to right). Each one of the six panels comprises a top view (center) and two lateral views (right and bottom). The top view shows a 3D traction map, where yellow vectors represent the in plane ($T_x$, $T_y$) components and the color map represents the vertical component $T_z$. The blue vectors in the lateral views show the tractions along two orthogonal bands passing through the center of the footprint. Scale bar, 50 μm. Scale vector, 50 Pa. **b** Inferred stresses on the free-standing cell layer of domes shown in **a**. The black arrows represent the two principal components of the inferred stress. The colormap represents the mean surface tension, $\sigma_I + \sigma_{II}$.

**c** Experimental vertical tractions at the dome-substrate interface for low, medium and high inflation levels. Left: low eccentricity, $n = 11$ (low), 16 (medium), 23 (high). Right: high eccentricity, $n = 15$ (low), 30 (medium), 16 (high). The tractions are shown after averaging in the four quadrants of the ellipses, taking advantage of the symmetry of the system. The polar angle $\beta$ along the footprint is defined in the top right schemes. **d** Vertical tractions inferred at the dome-substrate interface using cMSM for low, medium and high inflation levels. Left: low eccentricity, $n = 6$ (low), 9 (medium), 8 (high). Right: high eccentricity, $n = 8$ (low), 8 (medium), 8 (high). The line and shaded area indicate median and 95% CI of the median by bootstrapping (methods). Source data are provided as a Source Data file.

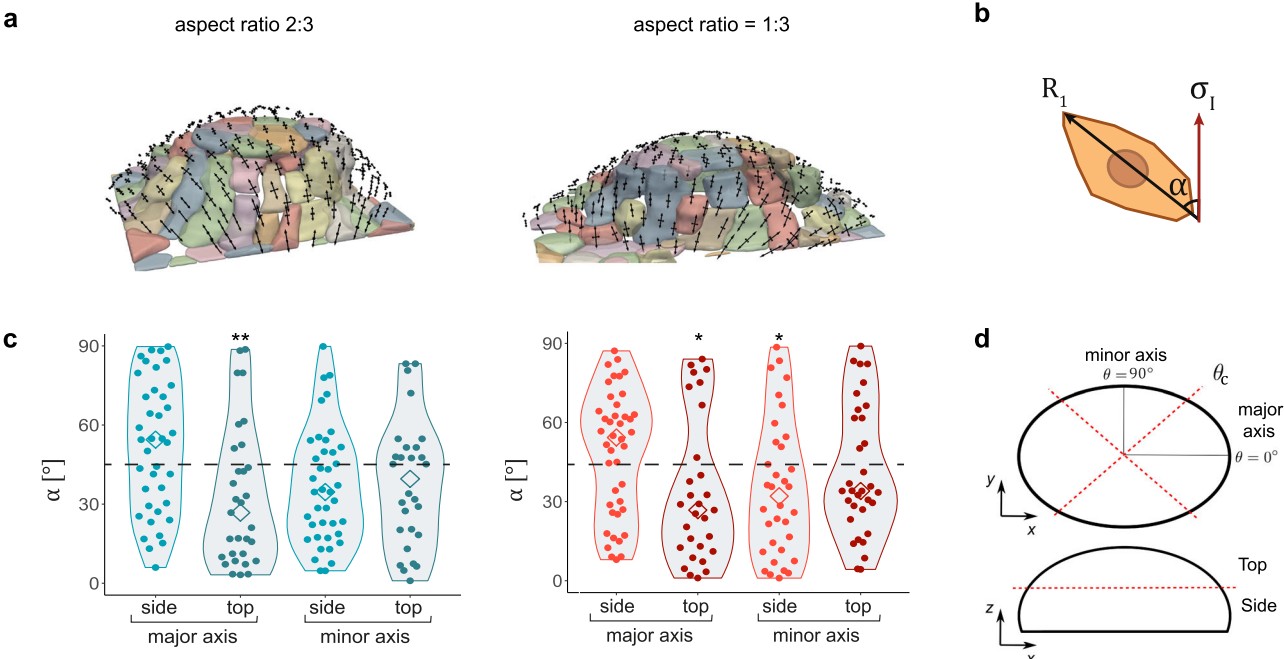

**Fig. 4 | Relationship between stress and cell orientation in elliptical domes.**
**a** Segmentation images of high inflation ellipsoidal domes with 2:3 (left) and 1:3 (right) aspect ratio. The black arrows represent the two principal components of the inferred stress. **b** Schematic showing the calculation of angle $\alpha$ between the longest cell axis ($R_I$) and the direction of the maximum principal stress ($\sigma_I$). **c** Distribution of angles $\alpha$ for the different regions of highly inflated domes (see "Methods" for significance test). $n = 39$ (major side, left), 31 (major top, left), 40 (minor side, left), 28 (minor top, left), 42 (major side, right), 29 (major top, right), 34 (minor side, right), and 33 (minor top, right). *$P < 0.05$ and **$P < 0.01$. $P = 7.6$E-03 (left panel, top major axis), $P = 0.010$ (right panel, top major axis), $P = 0.044$ (side minor axis). Diamond symbols indicate the median of the distributions. **d** Schematic indicating the different dome regions in which cells were binned. $\theta_c = 45°$ (2:3 aspect ratio) and 30° (1:3 aspect ratio). Top-Side threshold = 50% (2:3 aspect ratio) and 40% (1:3 aspect ratio) of dome height. Source data are provided as a Source Data file.

with normal tractions at the contact line between the free-standing epithelium and the substrate. For rectangular footprints, we found that normal tractions were higher along the long axis of the rectangle. For elliptical footprints, we found a close match between inferred stress and normal traction all along the contact line and as a function of local footprint curvature. Moreover, normal tractions increased with inflation, in agreement with inferred stresses. Together, these data provide a validation of our method showing that a direct measurement of monolayer shape and luminal pressure, together with the equations of mechanical equilibrium, suffice to infer the 2D stress tensor on a curved monolayer.

Using our validated technology, we probed the alignment between the longest cell axis and the direction of maximum principal stress. Such alignment has been observed in flat cell monolayers, both in vivo and in vitro, and it is a core prediction of stress inference methods based on cell shape[36]. In our elliptical domes, we found a significant alignment only in the regions of low curvature anisotropy (Fig. 4), such as the lateral side of the domes parallel to the longest axis of the ellipse footprint. By contrast, in the regions of highest curvature anisotropy the trend was opposite. One potential explanation for this

behavior is that cell alignment is determined by a competition between cell stretching and bending. In regions of low curvature anisotropy, bending is negligible and cells align in the direction of maximum principal stress. However, in regions of high curvature anisotropy, aligning in the direction of maximum stress would also imply bending of the cell body. In this mechanical scenario, it will be more favorable for cells to align in the direction of lowest curvature rather than in the direction of maximum stress. Similar arguments have been proposed to explain the alignment of adherent cells on cylindrical wires[31]. In the case of suspended domes lacking extracellular matrix, the subcellular mechanisms driving cell alignment likely involve changes in cell–cell adhesions and the cytoskeleton. Further studies should address this problem controlling not only curvature but also strain magnitude, rate, and history.

The shape of epithelia in vivo is often influenced by the presence of a basement membrane, by adjacent mesenchymal cells and smooth muscle, and by local bending moments and buckling instabilities[40–43]. Here we designed an approach that purposely ignores each of these confounding factors in order to study the behavior of the isolated free-standing monolayer under tight mechanical control. Additional

mechanical elements such as localized bending moments can be readily added to the system through optogenetic approaches[44–46] or controlled deposition of extracellular matrix[47]. Through the approach presented here, fundamental questions of how epithelial shape and stress anisotropies influence epithelial functions such as division[48,49], extrusion[50,51], intercalation[52] and stemness[13] can now be addressed quantitatively. The relations between shape, pressure and stress identified here can be used for the rational design of organoids and organ-on-a-chip systems based on epithelial layers[53,54].

## Methods

### Fabrication of soft silicone gels

Soft silicone gels were prepared using a protocol based on previous publications[55–58]. Briefly, the silicone elastomer was synthesized by mixing a 1:1 weight ratio of CY52-276A and CY52-276B poly-dimethylsiloxane (Dow Corning Toray). After degassing for 30 min in ice, the gel was spin-coated on glass-bottom dishes (35-mm, no. 0 coverslip thickness, Mattek) for 90 s at 400 r.p.m. The samples were then cured at 65 °C overnight. The substrates were kept in a clean and dry environment and they were used within 8 weeks of fabrication.

### Coating the soft PDMS substrate with fluorescent beads

After curing, a thin PDMS stencil with an inner diameter of 7 mm was placed on top of the soft PDMS gels. The region in the stencil was treated with (3-aminopropyl)triethoxysilane (APTES, Sigma-Aldrich, cat. no. A3648) diluted at 5% in absolute ethanol for 3 min, rinsed 3 times with ethanol absolute and rinsed once with type 1 water. Samples were incubated for 1 h with a filtered and sonicated solution of red fluorescent carboxylate-modified beads (FluoSpheres, Invitrogen) of 100 nm (220 nm filter) or 200 nm diameter (450 nm filter) in sodium tetraborate (3.8 mg/ml, Sigma-Aldrich) and boric acid (5 mg/ml, Sigma-Aldrich). Next, gels were rinsed 3 times with type 1 water.

### Passivation of soft PDMS substrates

After coating, soft PDMS gels were incubated with a solution of 1% poly-L-lysine (P2636, Sigma-Aldrich) diluted in type 1 water for 1 h. Next, the gels were rinsed 4 times with a 10 mM HEPES (1 M, Sigma-Aldrich) solution with pH in the range 8.2–8.4 and incubated with a 50 mg/ml dilution of PEG-SVA (Laysan Bio) in this pH-adjusted 10 mM HEPES for 1 h. Then, samples were rinsed 4 times with type I water and stored at 4 °C with type I water until photopatterning. All samples were used within 48 h of passivation.

### Protein photopatterning on soft PDMS

Photopatterning of soft PDMS gels was performed using the PRIMO optical module[59] (Alvéole) controlled by the Leonardo plugin (Alvéole) mounted on a Nikon inverted microscope (Nikon Instruments) equipped with a Super Plan Fluor ×20 ELWD lens (Nikon) and a DMD-based UV (375 nm). Before starting, the liquid on each sample was carefully aspirated (without letting the sample dry) and the sample was covered with PLPP photoactivator (Alvéole). The desired patterns for photoillumination were created using Inkscape (Inkscape Project) and loaded into Leonardo. The UV dose of all samples was set to 900 mJ/mm$^2$. After photopatterning, samples were rinsed 4 times with phosphate-buffered saline (PBS, Sigma-Aldrich), incubated for 5 min with a 0.02% fibronectin (F0895, Sigma-Aldrich) solution in PBS and rinsed thoroughly 5 times with PBS. For timelapse imaging experiments, filtered (220 nm filter) Alexa Fluor 647 conjugated fibrinogen was added to the fibronectin solution to allow visualization of the micropattern. Samples were stored at 4 °C until use (less than 48 h).

### Soft PDMS stiffness measurements

Gel stiffness was measured as previously described[14]. Briefly, a large 1mm-diameter metal sphere of known mass was used to generate an indentation on the non-photopatterned gels. Then, the depth of the indentation was quantified using confocal microscopy. Using the indentation depth and the sphere mass, we computed Young's modulus by applying Hertz theory and correcting for the finite thickness of the gel[60]. The resulting Young's modulus was 2.9 ± 0.7 kPa (mean ± SD, $n = 16$). Repeating the same measurements on photopatterned gels shows that stiffness was not affected by photoillumination (3.1 ± 0.8 kPa, mean ± SD, $n = 15$, Supplementary Fig. 10).

### Cell seeding

Before cell seeding, soft PDMS gels were exposed to sterilizing UV light for 15 min inside of the culture hood and then incubated with cell medium for 10 min. The medium was removed and 50 µl of new medium containing ~180,000 cells were placed inside of the stencil. Fifty minutes after seeding, the samples were washed using PBS to remove the non-attached cells and 1–2 ml of medium were added. Cells were seeded between 20 and 60 h before experiments (depending on dome size).

### Cell density calculation

To quantify cells in the dome, maximum projections of domes were obtained using Fiji (ImageJ 1.53t)[61]. Then the number of cells in the dome ($C_{in}$) and of those partially in of the dome but in contact with the substrate ($C_{bound}$) were manually counted for each dome ($n = 8$ for each dome size). To compute cell density, the following formula was used: Density = $(C_{in} + C_{bound}/2)/A$, where $A$ is the 2D area of the dome footprint.

### Three-dimensional traction microscopy

Three-dimensional traction forces were computed using traction microscopy with finite gel thickness[62,63]. The fluorescent beads coating the gel surface were imaged using 12 µm-thick confocal stacks with a z-step of 0.2 µm. Images of every experimental timepoint were compared to a reference image obtained after cell trypsinization. From these images, 3D displacement fields of the top layer of the gel were computed using home-made particle imaging velocimetry software, based on an iterative algorithm with a dynamic interrogation window size and implementing convergence criteria based on image intensity as described in previous publications[14,64].

### Pressure measurement

Pressure was measured by averaging normal traction over a central area of the dome to minimize boundary effects. The size of these regions depended on the geometry of the footprint.

### Curvature anisotropy

Curvature anisotropy was computed as $(1- k_{min}/k_{max})$, where $k_{min}$ and $k_{max}$ are the minimum and maximum curvature at each point of the meshed dome surface, respectively.

### Cell culture

All experiments were performed using a MDCK strain II line (kindly provided by Guillaume Charras) expressing CIBN−GFP−CAAX to visualize the plasma membrane. Cells were cultured in minimum essential medium with Earle's Salts and l-glutamine (Gibco) supplemented with 10% v/v fetal bovine serum (FBS; Gibco), 100 µg/ml penicillin and 100 µg/ml streptomycin. Cells were maintained at 37 °C in a humidified atmosphere with 5% $CO_2$. The cell line was obtained by viral infection of CIBN−GFP−CAAX. The cell line tested negative for mycoplasma contamination.

### Inhibition of contractility

Inhibition of contractility was performed as in ref. 14. In brief, time lapse imaging of confocal z-stacks was performed for 100 min every 20 min. After recording 60 min of baseline, Y27632 (30 µM dissolved in water) was added to wells and two more time points were measured.

## Time-lapse microscopy

Multidimensional acquisition for traction force measurements was performed using an inverted Nikon microscope with a spinning disk confocal unit (CSU-W1, Yokogawa), Zyla sCMOS camera (Andor, image size 2048 × 2048 pixels) using a 40 × 0.75 NA air lens (Nikon). The microscope was equipped with temperature and $CO_2$ control and controlled using Micro-manager (1.4.23) software[65].

## Areal strain calculation

Areal strain of the spherical domes was computed as $\varepsilon_a = (h/R_b)^2$ where $h$ is dome height and $R_b$ is the footprint radius.

## Estimation of vertical traction from stress inference

We calculated $l_T$ as $l_T = \tilde{\sigma}_{rz}/\tilde{T}_{rz}$, where $\tilde{\sigma}_{rz}$ is the median tension at the boundary for rectangular domes and $\tilde{T}_{rz}$ is the median experimental traction of rectangular domes. We obtained $l_T = 15.7$ μm.

## Dome shape segmentation

The luminal surface of domes was extracted by fitting a smooth surface to the point cloud representing the basal cell faces (Supplementary Note 2). The three-dimensional point cloud was obtained by manually adding points to the x-y, y-z, and x-z slices from the image stack using a custom Matlab (R2018b) code. The basal footprint was segmented first using x-y slices. Then, y-z and x-z slices spaced about 7.5 μm apart were processed.

## Cell segmentation

Cellular segmentation was carried out with Cellpose, a Python-based cell segmentation plugin[66]. For each z plane, cells were automatically segmented using a diameter of 100 pixels and then manually corrected. 2D segmentations from all planes were stitched together and cellular shape was extracted using home-made Matlab software.

## Statistical analysis

Comparisons between each group of unpaired samples were computed using the unpaired two-sided Wilcoxon rank sum test. Comparisons between groups of paired samples (long vs short sides in rectangular domes) were computed using the paired two-sided Wilcoxon rank sum test. For domes with elliptical footprint, 95% confidence intervals were computed by bootstrapping of the median with $10^4$ replicates, using the "boot" function in R. For cell orientation data, we used Matlab to generate $10^4$ uniform distributions with the size of the experimental data set. We then computed the median of each distribution and the distribution of medians. The position of the experimental median in this distribution defines the provided P-value. Statistical analysis was performed with RStudio version 1.4.1717 (running R version 3.6.3).

## Reporting summary

Further information on research design is available in the Nature Portfolio Reporting Summary linked to this article.

# Data availability

Source data are provided in this paper.

# Code availability

The cMSM code along with experimental datasets are included in this submission and can be downloaded from Zenodo [https://zenodo.org/record/7921052].

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

## Acknowledgements

We thank C. Pérez-González, N. Castro, and all of the members of the Roca-Cusachs, Arroyo, and Trepat laboratories for their discussions and support. This work was supported by: Generalitat de Catalunya (Agaur, SGR-2021-01425 to X.T., SGR-2021-00523 to R.S., the CERCA Programme, and "ICREA Academia" award to M.A. and P.R-C.); Spanish Ministry for Science and Innovation MICCINN/FEDER (PID2021-128635NB-I00, MCIN/AEI/ 10.13039/501100011033 and "ERDF-EU A way of making Europe" to X.T., PID2019-110949GB-I00 to M.A., PID2019-110298GB-I00 to P.R-C., PID2021-128674OB-I00, RTI2018-101256-J-I00, and RYC2019-026721-I to R.S.); European Research Council (Adv-883739 to X.T., CoG-681434 to M.A.); Fundació la Marató de TV3 (project 201903-30-31-32 to X.T.); Deutsche Forschungsgemeinschaft (DFG GO3403/1-1 to T.G.); IBEC, IRB, and CIMNE are recipients of a Severo Ochoa Award of Excellence from the MINECO; European Commission (H2020-FETPROACT-01-2016-731957 to P.R-C.); La Caixa Foundation (LCF/PR/HR20/52400004 and ID 100010434 under the agreement LCF/PR/HR20/52400004 to P.R-C. and X.T.). R.S. is a Serra Húnter fellow.

## Author contributions

A.M.-L., M.A., and X.T. conceived the project. A.M.-L. performed experiments and analyzed data. A.O. and A.T.-S. developed software and performed simulations. S.K. and M.A. conceived and implemented the force inference method. R.S. developed tools for surface micro-patterning and statistical methods. E.L. and M.G.-G. developed software for traction calculation and image analysis. T.G. performed myosin inhibition experiments. P.R-C. contributed technical expertize, materials, and discussion. A.M.-L., S.K., A.O., M.A., and X.T. wrote the manuscript. S.K., M.A., and X.T. supervised the project.

## Competing interests

The authors declare no competing interests.
