## [Peer Review File · Nature Communications]

Mapping mechanical stress in curved epithelia of designed size and shapeREVIEWER COMMENTS

Reviewer #1 (Remarks to the Author):

In this study, Marin-Llaurado et al. designed curved epithelial monolayers of varying size to determine the correspondence between shape and mechanical stress in epithelial layers independently of underlying material properties. In particular, they managed to create anisotropic shapes and observed that cells align with the direction of maximum principal stress.

Overall, the authors provide an interesting approach based on the combination of experimental and computational tools to vary epithelium geometries and map stress. This work could be of help in understanding the formation and regulation of 3D epithelia, which is of interest for a broad scientific readership. However, I found the paper too preliminary for some aspects and I recommend to clarify and strengthen some of the experiments presented here before publication.

1- The significance of data presented in figures 2 and 4 should be better clarified, since some conclusions are based on figure 2d, where the significance for different results is different and it is unclear what are the determinants for the increase in the central and right plots. I find also very difficult to understand Figure 4c and the impact of measurement distribution on the final interpretation for cell orientation.

2- Methods: a) could the authors clarify how they quantify high and low fibronectin coatings and whether matrix surface density is homogeneous at the margin of the pattern (usually matrix proteins tend to accumulate at the boundary). b) have the authors checked the hydrogels after the 15 min UV exposure prior to cell seeding? Elastomers tend to get very brittle and change their surface properties.

3- I am missing completely the molecular mechanism explaining the cell alignment. The authors performed the cell experiment using a line expressing a fluorescent construct to visualize the plasma membrane, however there are no information about the relation e.g. between cell orientation, adherens junction regulation and actin (I refer to Figure 4). Also, in Figure 4a, in the segmentation there seem to be even no consistent and accurate representation of cell-cell contacts.

Reviewer #2 (Remarks to the Author):

This is an excellent manuscript, which combined experiments and a novel computational tool to infer stress distributions in inflated epithelia with different geometrical footprints. This framework goes significantly beyond the previous work in the literature that inferred stresses only in spherical epithelia. The provided computational tool will also be a great resource for the community. The manuscript is very clearly written, and all claims are supported by evidence. I recommend publication after the following clarifications are made.

1.) Normal tractions showed a close match with the inferred stress along the contact line. Would it be possible to also compare the measured tractions in the x-y plane with the inferred stresses? I can imagine that such comparison is difficult because the neighboring cells attached to the substrate are also pulling/pushing on the cells at the contact line.

2.) In most regions, there was a tendency of cells to align with the direction of the maximum principal stress. How does this correlation depend on the magnitude of the maximum principal stress? Are cells more aligned for higher values of the maximum principal stress?

3.) How was the luminal pressure measured in experiments? Was it measured by averaging normal tractions over some regions or were normal tractions measured at specific locations (e.g., at the center)?

- 4.) What surface tension is plotted in Fig. 1e? Does each point correspond to average stress over the whole epithelia or does it correspond to local stress at one of the points on the epithelia?
- 5.) How are different levels of inflation defined (high/low in Fig. 2, high/medium/low in Fig. 3)? Is there some quantitative metric that was used to bin the data?
- 6.) Add descriptions for the hydrostatic stress color maps also in the captions of Figs. 2b and 3b.
- 7.) In captions of Fig. 3c,d, clarify that the polar angle is measured relative to the long axis of the ellipse.
- 8.) Clarify what the diamond symbols represent in Fig. 4c. I guess they correspond to the averages of the distribution.
- 9.) In methods explain what are the parameters 'h' and 'a' in the areal strain calculation. For the consistency with supplements 'a' should be renamed to 'R_b'
- 10.) The symbols for the curvature tensor and tractions in Extended Data Figure 4 should be made consistent with the notation in the Supplementary Materials (SM).
- 11.) In Section 1.5 of the SM, explain how the effective cortical viscosity was implemented.
- 12.) In Section 2.4 of the SM, explain how the initial surface triangulation of the experimental point cloud was done. In 2D one can use standard Delaunay triangulation algorithms, but it is not clear how to do this easily for points in 3D.
- 13.) In Figs. 5e, 7e, and 8e in the SM, also add green arrows indicating the expected solutions. Similarly, to how it was done for Figs. 4, 9e, 10e, 11e

Reviewer #3 (Remarks to the Author):

In their manuscript entitled "Mapping mechanical stress in curved epithelia of designed size and shape", Marín-Llauradó et al. present cMSM, a novel computational method to infer stresses in curved epithelia. For non-spherical epithelial lumens, surface stresses cannot be easily calculated or measured due to the anisotropic nature of stress distribution. The computational approach presented here allows the inference of stress tensors on the surface of curved epithelia of any shape, solely based on their geometry and luminal pressure. As a model of epithelial lumens, the authors grow doming MDCK epithelia with various footprint shapes, in which luminal pressure can be measured using traction force microscopy. They use this method to map the surface stress on non-spherical epithelial domes with a rectangular or ellipsoidal footprint. As expected, stress tensors along the epithelial surface are anisotropic. They validate their computational approach by comparing experimentally accessible traction forces at the cell-substrate interface with their computed values at these locations and find a good correspondence. Finally, they ask if cellular long axes align with the direction of anisotropic stress tensors in non-spherical epithelia, which they approach by cell segmentation. However, they do not find a global correlation of these parameters.

The manuscript is nicely written and the figures are clear. The analysis presented in this paper will be a useful tool for researchers interested in epithelial mechanics. However, due to the lack of an application of the method beyond the synthetic MDCK domes, biological conclusions are limited. Since we lack the relevant expertise, we cannot comment on the validity of the underlying physical assumptions, computational methods and data evaluations. On the experimental side of the paper, experiments consists in a simple and elegant way to create epithelia of controlled shapes and sizes.

The authors present few interesting conclusions about epithelial mechanics, such as the correlation of dome diameter and lumen pressure, the absence of mechanosensitive response in this range of lumen sizes and pressure, as well as the non-globality of cell alignment of cell axes with stress orientation.

Specific comments:

- Fig1d: epithelium curvature seems to scale linearly with lumen pressure (before a plateau for very small lumen of just a few cells, which may be difficult to characterize). This scaling is based on 3 different lumen sizes and the data are quite scattered around the mean values. Could larger domes be made to extend those measurements beyond what is currently explored?
- The authors conclude that different properties of apical / basal cell surface are negligible when considering forces in the epithelial layer. However, in biological contexts, lumens are often embedded in the ECM / underlying cells with their basal side. How will this be reflected in the cMSM approach?
- To further validate cMSM, the authors could alter the mechanical properties of epithelial sheets by myosin, actin or microtubule manipulation and see how good their method would be at predicting stresses in these conditions.
- The authors find no global correlation of cell axis alignment with the main stress tensor direction. Which other factors could influence cell orientation?
- References 5 and 6 about the developing otic vesicle concerns zebrafish and not drosophila
- Fig2d: there is "side x" overlaid on the (many) asterisks indicating statistical significance of the middle graph.

REVIEWER COMMENTS

Reviewer #1 (Remarks to the Author):

We thank the reviewers for their comments and suggestions, which have greatly improved the quality of our paper. We hope the reviewed version provides a clearer and more comprehensive presentation of our findings. Our point-by-point response to their comments is below.

In this study, Marin-Llaurado et al. designed curved epithelial monolayers of varying size to determine the correspondence between shape and mechanical stress in epithelial layers independently of underlying material properties. In particular, they managed to create anisotropic shapes and observed that cells align with the direction of maximum principal stress.

Overall, the authors provide an interesting approach based on the combination of experimental and computational tools to vary epithelium geometries and map stress. This work could be of help in understanding the formation and regulation of 3D epithelia, which is of interest for a broad scientific readership. However, I found the paper too preliminary for some aspects and I recommend to clarify and strengthen some of the experiments presented here before publication.

We thank the reviewer for the positive comments on our manuscript and for the constructive criticism and suggestions.

1- The significance of data presented in figures 2 and 4 should be better clarified, since some conclusions are based on figure 2d, where the significance for different results is different and it is unclear what are the determinants for the increase in the central and right plots. I find also very difficult to understand Figure 4c and the impact of measurement distribution on the final interpretation for cell orientation.

From the reviewer's comment we now see that our description of Fig. 2d was unclear. There are two determinants for the increase in the vertical component of the traction. The first one is inflation: increasing inflation always increases the normal traction regardless of the shape of the footprint. The second one is the stress anisotropy in the monolayer, which causes the normal traction to be higher along the longest side of the footprint than along the shortest one. Since both determinants of the traction increase are relevant to validate our technique, we tested the statistical significance of both independently using Wilcoxon rank sum test for paired or unpaired samples depending on the comparison. Hence our plotting of statistical significance in the central and right panels of Fig. 2d. We have now re-written the description of this figure to clarify this point (p. 5, second paragraph).

In Fig. 4, we assessed significance as follows. We generated 10^4 uniform distributions, each with the experimental sample size. We then computed the median of each distribution and the distribution of the medians. Finally, we assessed the p-values by locating the experimental median in this simulated distribution. We clarify this test in the methods section (Statistical Analysis).

2- Methods: a) could the authors clarify how they quantify high and low fibronectin coatings and whether matrix surface density is homogeneous at the margin of the pattern (usually matrix proteins tend to accumulate at the boundary). b) have the authors checked the

hydrogels after the 15 min UV exposure prior to cell seeding? Elastomers tend to get very brittle and change their surface properties.

The reviewer raises two relevant issues in surface photopatterning. Regarding point a) we now provide a representative fluorescence image of the photopatterned gels and the quantification of fluorescence levels for several footprints. These are presented in new Extended Data Figure 1. These data show the absence of systematic accumulation of patterned ECM at the boundary of the pattern. Regarding point b) we used the ball indentation method to quantify substrate stiffness with and without photopatterning, with no significant difference between both conditions. These data are shown in Extended Data Figure 10 and mentioned in the methods section (Soft PDMS stiffness measurements).

3- I am missing completely the molecular mechanism explaining the cell alignment. The authors performed the cell experiment using a line expressing a fluorescent construct to visualize the plasma membrane, however there are no information about the relation e.g. between cell orientation, adherens junction regulation and actin (I refer to Figure 4). Also, in Figure 4a, in the segmentation there seem to be even no consistent and accurate representation of cell-cell contacts.

We agree with the referee on the importance of understanding the mechanisms of cell alignment in curved monolayers. We now provide additional data and discussion on those mechanisms, but their description in molecular detail is beyond the scope of our paper. In this regard, we would like to point out that even in the simplest case of a single cell or a monolayer on a flat 2D substrate, mechanisms of cell alignment with stress remain poorly understood. Our goal in this study was to provide a methodological approach to begin to address this problem with high control of geometry and full mapping of the stress tensor. Our data on cell alignment are a proof of concept of the technique and we expect to address mechanisms of cell alignment in the future. We note that identifying such mechanisms with molecular detail will require not only mapping stress and curvature accurately, as our approach enables, but also to control the magnitude, rate and strain history of the dome, which is currently unavailable in our system. We are developing new approaches to provide such control by imposing hydraulic pressure to the dome underside.

However, we agree with the referee that our original submission required additional evidence and discussion to clarify potential mechanisms of cell alignment. We now provide additional data and discussion on the possibility that besides aligning in the direction of maximum principal stress, cells might also align in the direction of minimum curvature in order to minimize bending. This might explain why in the regions of highest curvature of the domes, cells tend to orient perpendicular rather than parallel to maximum principal stress. New data on curvature measurements can be found in Extended Data Fig. 9. These data are presented on p. 5 (last paragraph) and discussed on p. 7 (second paragraph).

We agree with the reviewer that our segmentation strategy does not capture accurately cell-cell junctions. For the purpose of our study, the main goal of our segmentation approach was the identification of the 3D axes of orientation of the cell body, which does not require accurate information on cell-cell junctions.

Reviewer #2 (Remarks to the Author):

This is an excellent manuscript, which combined experiments and a novel computational tool to infer stress distributions in inflated epithelia with different geometrical footprints. This framework goes significantly beyond the previous work in the literature that inferred stresses only in spherical epithelia. The provided computational tool will also be a great resource for the community. The manuscript is very clearly written, and all claims are supported by evidence. I recommend publication after the following clarifications are made.

We thank the reviewer for the positive comments on our work and for the detailed critique.

1.) Normal tractions showed a close match with the inferred stress along the contact line. Would it be possible to also compare the measured tractions in the x-y plane with the inferred stresses? I can imagine that such comparison is difficult because the neighboring cells attached to the substrate are also pulling/pushing on the cells at the contact line.

The referee's intuition is correct. At the contact line, the x-y tractions have two contributions (mentioned at the end of the second paragraph of the Results section). The first one is the pulling force exerted by the suspended monolayer. The second one is the pulling force exerted by the adhered monolayer. As a result, the traction vector at the contact line is generally not tangential to the suspended monolayer and the estimation of x-y tractions is not straightforward.

2.) In most regions, there was a tendency of cells to align with the direction of the maximum principal stress. How does this correlation depend on the magnitude of the maximum principal stress? Are cells more aligned for higher values of the maximum principal stress?

We tested whether the alignment angle was correlated with the maximum principal stress for each of the four regions of the domes (new Extended Data Fig. 8, mentioned on p. 5, last paragraph). We did not find a correlation. As an alternative, we tested whether the alignment angle was correlated with stress anisotropy, expressed as the ratio between minimal and maximal principal stresses. We did not find a correlation either (also shown in Extended Data Fig. 8). To a large extent we attribute this variability to the high scatter in the angular distribution and possibly to the existence of two competing mechanisms for cell alignment associated to the energy costs of stretching vs bending. We discuss this possibility on p. 5 (last paragraph) and on p. 7 (second paragraph).

3.) How was the luminal pressure measured in experiments? Was it measured by averaging normal tractions over some regions or were normal tractions measured at specific locations (e.g., at the center)?

Pressure was measured by averaging normal traction over a central area of the dome to minimize boundary effects. The size and shape of this area depended on the geometry of the footprint. We now mention this methodological aspect in the methods section (Pressure measurement).

4.) What surface tension is plotted in Fig. 1e? Does each point correspond to average stress over the whole epithelia or does it correspond to local stress at one of the points on the epithelia?

Each point corresponds to surface tension computed from Laplace's law using the measured pressure and radius. We now clarify this point in the figure caption.

5.) How are different levels of inflation defined (high/low in Fig. 2, high/medium/low in Fig. 3)? Is there some quantitative metric that was used to bin the data?

Data were binned according to arbitrary height thresholds to ensure a sufficient number of data points in each category. Results did not depend on the specific value of the threshold.

6.) Add descriptions for the hydrostatic stress color maps also in the captions of Figs. 2b and 3b.

Thanks, we added the descriptions to the captions.

7.) In captions of Fig. 3c,d, clarify that the polar angle is measured relative to the long axis of the ellipse.

We added the requested clarification in the figure caption as well as a cartoon in the figure illustrating the definition of the angle.

8.) Clarify what the diamond symbols represent in Fig. 4c. I guess they correspond to the averages of the distribution.

They represent the median. We now clarify this in the caption.

9.) In methods explain what are the parameters 'h' and 'a' in the areal strain calculation. For the consistency with supplements 'a' should be renamed to 'R_b'

We agree and have unified the nomenclature with the supplement.

10.) The symbols for the curvature tensor and tractions in Extended Data Figure 4 should be made consistent with the notation in the Supplementary Materials (SM).

We agree and have unified notation.

11.) In Section 1.5 of the SM, explain how the effective cortical viscosity was implemented.

We added the underlined text to the supplement.

However, as discussed in (Perez-Gonzalez et al. Nat Cell Biol, 2021), solving Eq. (4) leads to uncontrolled mesh distortion as nodes can move tangentially without changing cell area or volume. To avoid such tangential motions of nodes, we added an effective cortical viscosity, which vanishes at equilibrium to avoid biasing the end results. Taking a similar approach to (Ma and Klug, J Comput Phys, 2008), we treated cell cortices as hyperelastic surfaces deforming with respect to an evolving reference configuration. However, instead of an iterative update, we considered the evolution rule of the reference configuration towards the current configuration to be orders of magnitude faster than the dome inflation process, making all stored elastic energy negligible at each increment of the dome inflation process. Therefore, the dome inflation process can be seen as the quasi-static evolution of active-viscous cortices with fixed cellular volume and an increasing lumen volume.

12.) In Section 2.4 of the SM, explain how the initial surface triangulation of the experimental point cloud was done. In 2D one can use standard Delaunay triangulation algorithms, but it is not clear how to do this easily for points in 3D.

The initial surface triangulation of the experimental 3D point cloud was obtained using an implementation of the crust algorithm for surface reconstruction from unorganized 3D sample points for open surfaces available at:

<https://www.mathworks.com/matlabcentral/fileexchange/63731-surface-reconstruction-from-scattered-points-cloud-open-surfaces>.

An explanation and reference to this open-source package is added to the discussion in Section 2.4 of the Supplementary material.

13.) In Figs. 5e, 7e, and 8e in the SM, also add green arrows indicating the expected solutions. Similarly, to how it was done for Figs. 4, 9e, 10e, 11e

Figs. 5e, 7e, and 8e have been updated to include the green arrows for visual comparison of the expected and obtained solutions. Note that for the spherical cap in Fig. 5e, the mismatch between directions of green and black arrows is anticipated since the surface tension is isotropic for which the orientation of the principal directions is arbitrary.

Reviewer #3 (Remarks to the Author):

In their manuscript entitled “Mapping mechanical stress in curved epithelia of designed size and shape”, Marín-Llauradó et al. present cMSM, a novel computational method to infer stresses in curved epithelia. For non-spherical epithelial lumens, surface stresses cannot be easily calculated or measured due to the anisotropic nature of stress distribution. The computational approach presented here allows the inference of stress tensors on the surface of curved epithelia of any shape, solely based on their geometry and luminal pressure. As a model of epithelial lumens, the authors grow doming MDCK epithelia with various footprint shapes, in which luminal pressure can be measured using traction force microscopy. They use this method to map the surface stress on non-spherical epithelial domes with a rectangular or ellipsoidal footprint. As expected, stress tensors along the epithelial surface are anisotropic. They validate their computational approach by comparing experimentally accessible traction forces at the cell-substrate interface with their computed values at these locations and find a good correspondence. Finally, they ask if cellular long axes align with the direction of anisotropic stress tensors in non-spherical epithelia, which they approach by cell segmentation. However, they do not find a global correlation of these parameters.

The manuscript is nicely written and the figures are clear. The analysis presented in this paper will be a useful tool for researchers interested in epithelial mechanics. However, due to the lack of an application of the method beyond the synthetic MDCK domes, biological conclusions are limited. Since we lack the relevant expertise, we cannot comment on the validity of the underlying physical assumptions, computational methods and data evaluations. On the experimental side of the paper, experiments consists in a simple and elegant way to create epithelia of controlled shapes and sizes. The authors present few interesting conclusions about epithelial mechanics, such as the correlation of dome diameter and lumen pressure, the absence of mechanosensitive respond in this range of lumen sizes and pressure, as well as the non-globality of cell alignment of cell axes with stress orientation.

We thank the reviewer for the positive assessment of our work and for the detailed and constructive critique.

Specific comments:

- Fig1d: epithelium curvature seems to scale linearly with lumen pressure (before a plateau for very small lumen of just a few cells, which may be difficult to characterize). This scaling is based on 3 different lumen sizes and the data are quite scattered around the mean values. Could larger domes be made to extend those measurements beyond what is currently explored?

Unfortunately, larger domes cannot be made with our current technology. We rely on the ability of domes to pump osmolytes across their surface to delaminate and inflate. We found that beyond 200 μm in diameter (the maximum size shown in the manuscript), the monolayer does not fully delaminate. We are currently working on new methods to delaminate domes by directly imposing pressure differences, but these methods do not allow for traction measurements and are not yet ready for publication.

- The authors conclude that different properties of apical / basal cell surface are negligible when considering forces in the epithelial layer. However, in biological contexts, lumens are often embedded in the ECM / underlying cells with their basal side. How will this be reflected in the cMSM approach?

This interesting point highlights an assumption embedded in the presented cMSM formulation that the membrane supports a uniform luminal pressure. If this is not the case, for instance when the pressurized lumen is supported by ECM on one side, we need to account for the contact traction forces between ECM and the lumen in the tangential and normal force balance equations. The proposed cMSM approach can be readily modified to account for these additional external forces acting on the membrane. Surface stresses can be resolved in absolute terms if these contact traction forces can be measured in addition to the luminal pressure. This will amount to performing 3D TFM on the ECM in contact with the curved membrane. Although experimentally challenging, the proposed approach in principle can be extended to account for such a biological context. We have added a discussion on this point in Section 2.1 of the Supplementary Material.

- To further validate cMSM, the authors could alter the mechanical properties of epithelial sheets by myosin, actin or microtubule manipulation and see how good their method would be at predicting stresses in these conditions.

This is an excellent suggestion. To further validate our technique we treated ellipsoidal domes with ROCK inhibitor Y27632. Upon treatment with this drug, both dome tension and pressure showed a sudden decrease consistent with the impairment of myosin activity. We show this data in new Extended Data Fig. 7.

- The authors find no global correlation of cell axis alignment with the main stress tensor direction. Which other factors could influence cell orientation?

We thank the referee for raising this point, which has led us to rethink our mechanical analysis of cell orientation. On flat 2D surfaces, we showed in the past that cells in epithelial monolayers align and move in the direction of maximum principal stress (Tambe et al, *Nat Mater*, 2011). However, in curved monolayers there is an additional mechanical ingredient to take into account, which is the resistance of cells to bending (see for example the theory by Biton and Safran, *Phys Biol* 2009, to explain cell orientation on cylindrical wires). When coupling between bending and alignment dominates, then cells will tend to align in the

direction of lower curvature (lower bending) rather than in that of maximum stress. In our ellipsoidal domes, the directions of minimum curvature and maximum principal stress are roughly orthogonal, and curvature varies depending on the dome region, providing an interesting configuration to test the competing effects of stretching vs bending. Consistent with our reasoning, in the regions with lowest curvature anisotropy (those labelled as *minor axis top*, *minor axis side* and *major axis top* on Fig. 4d) cells were predominantly oriented in the direction of maximum stress. By contrast, in the region of highest curvature anisotropy (labelled as *major axis side*) cells tended to orient with the direction of minimal curvature (i.e. normal to maximum stress). We now discuss this point in the main manuscript. New data on curvature measurements can be found in Extended Data Fig. 9. These data are presented on p. 5 (last paragraph) and discussed on p. 7 (second paragraph). We plan to further test these ideas in future studies using devices in which the magnitude and rate of inflation as well as the mechanical history of the sample can be controlled.

- References 5 and 6 about the developing otic vesicle concerns zebrafish and not drosophila

Thanks for pointing out this ambiguity. Reference 5 is about the zebrafish otic vesicle and reference 6 is about the drosophila embryo. We have split the references and moved them earlier in the sentence to clarify this point.

- Fig2d: there is "side x" overlaid on the (many) asterisks indicating statistical significance of the middle graph.

Thanks for pointing out this mistake. It has now been corrected.

REVIEWERS' COMMENTS

Reviewer #1 (Remarks to the Author):

The authors have addressed my comments and the manuscript has significantly improved. I just have two minor points:

1. In the discussion, the authors have added a paragraph on the possible explanation for cell alignment. Could the authors elaborate on the involvement of cell cytoskeleton, and cell-cell interactions, considering that the domes are free-standing.
2. Fig 2d central panel has the text "side x" overlapping the significance symbols for the long group. Please remove this text.

Reviewer #2 (Remarks to the Author):

The authors have addressed all of my previous concerns, and I recommend publication.

Minor comments:

- 1.) The polar angle in Figure 3c is defined as beta, but the main text refers to the angle theta.
- 2.) The middle panel in Figure 2d has the text "side x" overlaid on top of the asterisks
- 3.) In the Extended Data Figure 8, the top and bottom rows of panels should be switched, otherwise the plots are not consistent with the captions.

Reviewer #3 (Remarks to the Author):

In their revised version, the authors have analyzed the stress and pressure of domes after treatment with Y27632. While this addition provides information about the potential applications of their powerful and interesting analysis, the way this experiment is currently presented is unfortunately not satisfactory.

Stresses and pressure are measured before and after the addition of Y27632. There is no description of that experiment in the methods: where is the drug coming from, what is the drug diluted into, at what concentration, are the domes treated with the vehicle before being treated with the drug (if that is DMSO, it could have an effect of its own)? At present, the reader cannot understand how the experiments were performed and they cannot be reproduced.

Also, Y27632 is presented as an inhibitor of actomyosin activity. Unfortunately, that is too often what is found in the literature but it is actually not that straightforward. Y27632 inhibits Rock, as mentioned by the authors, which can lead to reduced actomyosin activity. However, Y27632 is also a potent inhibitor of atypical protein kinase C (aPKC) (Atwood and Prehoda 2009), which is key to maintain polarized fluid transport in MDCK cells. Therefore, the measurements presented by the authors could be interpreted very differently if the effects of Y27632 were the result of aPKC inhibition: the pressure drop would result from decreased pumping rather than decreased actomyosin activity. This needs to be discussed and ideally a different perturbation would be helpful if the authors want to make a point of actomyosin being important for dome stress and luminal pressure. In addition, the authors could look into areas of the monolayer without domes: if tractions to the ECM were decreased after Y27632

treatment, that would be a good indication that actomyosin activity would be affected.

REVIEWER COMMENTS

Reviewer #1 (Remarks to the Author):

The authors have addressed my comments and the manuscript has significantly improved. I just have two minor points:

1. In the discussion, the authors have added a paragraph on the possible explanation for cell alignment. Could the authors elaborate on the involvement of cell cytoskeleton, and cell-cell interactions, considering that the domes are free-standing.

We now mention that in the absence of ECM, the subcellular mechanisms accounting for cell alignment likely reside in the cytoskeleton or cell-cell interactions (page 7, paragraph 2, last 4 lines).

2. Fig 2d central panel has the text "side x" overlapping the significance symbols for the long group. Please remove this text.

This text has now been removed.

Reviewer #2 (Remarks to the Author):

The authors have addressed all of my previous concerns, and I recommend publication.

Minor comments:

1.) The polar angle in Figure 3c is defined as beta, but the main text refers to the angle theta.

We have corrected the text. All angles are consistently called β .

2.) The middle panel in Figure 2d has the text "side x" overlaid on top of the asterisks

This text has been removed.

3.) In the Extended Data Figure 8, the top and bottom rows of panels should be switched, otherwise the plots are not consistent with the captions.

We have rewritten the legend to correct this error.

Reviewer #3 (Remarks to the Author):

In their revised version, the authors have analyzed the stress and pressure of domes after treatment with Y27632. While this addition provides information about the potential

applications of their powerful and interesting analysis, the way this experiment is currently presented is unfortunately not satisfactory.

Stresses and pressure are measured before and after the addition of Y27632. There is no description of that experiment in the methods: where is the drug coming from, what is the drug diluted into, at what concentration, are the domes treated with the vehicle before being treated with the drug (if that is DMSO, it could have an effect of its own)? At present, the reader cannot understand how the experiments were performed and they cannot be reproduced.

In the revised manuscript we provide the protocol used for Y27632 in the methods section. We note that this protocol is the same used in Latorre et al (Nature, 2018).

Also, Y27632 is presented as an inhibitor of actomyosin activity. Unfortunately, that is too often what is found in the literature but it is actually not that straightforward. Y27632 inhibits Rock, as mentioned by the authors, which can lead to reduced actomyosin activity. However, Y27632 is also a potent inhibitor of atypical protein kinase C (aPKC) (Atwood and Prehoda 2009), which is key to maintain polarized fluid transport in MDCK cells. Therefore, the measurements presented by the authors could be interpreted very differently if the effects of Y27632 were the result of aPKC inhibition: the pressure drop would result from decreased pumping rather than decreased actomyosin activity. This needs to be discussed and ideally a different perturbation would be helpful if the authors want to make a point of actomyosin being important for dome stress and luminal pressure. In addition, the authors could look into areas of the monolayer without domes: if tractions to the ECM were decreased after Y27632 treatment, that would be a good indication that actomyosin activity would be affected.

This is a great point. We now provide new evidence showing that the response to Y27632 arises from inhibition of actomyosin rather than an alteration in flows. First, we follow the reviewer's suggestion and show that in the areas of the monolayer without domes, Y27632 drives an acute drop in tractions, indicating a downregulation of actomyosin activity. Second, we now show that quickly after the addition of Y27632, tension in the dome drops whereas dome volume does not change. This result provides further support to our conclusion that Y27632 mainly affects actomyosin activity rather than transport within the time scale of our experiments. The new data are presented in Supplementary Fig. 7 (panels g and h) and discussed on page 5, paragraph 3.